# Whole-Body Cryotherapy Improves Asprosin Secretion and Insulin Sensitivity in Postmenopausal Women–Perspectives in the Management of Type 2 Diabetes

**DOI:** 10.3390/biom13111602

**Published:** 2023-10-31

**Authors:** Magdalena Wiecek, Jadwiga Szymura, Justyna Kusmierczyk, Maria Lipowska, Zbigniew Szygula

**Affiliations:** 1Department of Physiology and Biochemistry, University of Physical Education in Kraków, 31-571 Kraków, Poland; 2Department of Clinical Rehabilitation, University of Physical Education in Kraków, 31-571 Kraków, Poland; 3Laboratory of Biochemistry and Molecular Biology, University of Physical Education in Kraków, 31-571 Kraków, Poland; 4Department of Sports Medicine and Human Nutrition, University of Physical Education in Kraków, 31-571 Kraków, Poland

**Keywords:** cryostimulation, glucose intolerance, hormonal regulation, adipokines, menopause, diabetes therapy

## Abstract

Type 2 diabetes (T2DM) is a global problem. The effect of whole-body cryotherapy (WBC) on metabolism in humans is postulated. The aim of this study was to determine the effect of WBC on asprosin concentrations, glucose homeostasis and insulin resistance in postmenopausal women with T2DM. Changes in fasting blood glucose (FBG), glycated haemoglobin (HbA1c), insulin, asprosin, insulin-resistance indices (HOMA-IR, Quicki), the triglyceride–glucose index (TyG) and C-reactive protein (CRP) were determined. Determination was carried out after 30 WBCs (3 min, −120 °C), applied in six series of five treatments, with 2-day breaks in postmenopausal women with T2DM and the results were compared to changes in postmenopausal women without T2DM (CON). Blood was collected before 1 WBC (T0), after 30 WBCs (T1) and 2 weeks after their completion (T2). In the T2DM group, there was a significant decrease in FBG and HbA1c in T1 and T2, as well as a significant decrease in insulin, HOMA-IR and CRP, and an increase in the Quicki index in T2. In the CON group, the concentration of asprosin at T2 was significantly lower than at T0. There was a significantly positive correlation between asprosin and FBG and HOMA-IR, and a trend towards a decrease of asprosin concentration in T2 in postmenopausal women with T2DM.

## 1. Introduction

Diabetes is defined as a chronic metabolic disease manifested by hyperglycaemia [1]. The pathophysiological basis of this disease is impaired secretion of insulin and/or decreased sensitivity of cells to its action, including myocytes, cardiomyocytes, hepatocytes and adipocytes [2]. Chronic hyperglycaemia is associated with damage to or dysfunction of various organs, especially the eyes, kidneys, heart and blood vessels [3]. The most common form of this disease is type 2 diabetes mellitus (T2DM), which accounts for about 80–90% of all diabetes cases [4].

According to the report of the International Diabetes Federation (IDF), in 2021, 537 million adults (aged 20–79) suffered from diabetes worldwide, of which 6.7 million died from this illness. According to forecasts, the number of people with diabetes is expected to increase to 643 million by 2030 and to 783 million by 2045. In Europe, in 2045, a 13% increase in the number of diabetic patients is expected compared to the number in 2021. The significant increase in diabetes patients also refers to individuals above the age of 45 [5].

The development of metabolic complications in type 2 diabetes is associated, among others, with impaired secretion of adipocytokines. A correlation has been found between increased secretion of tumor necrosis factor α (TNF-α), interleukins 6 (IL-6), leptin and resistin, as well as decreased secretion of adiponectin and the development of insulin resistance [6].

Asprosin was discovered in recent years. It is a peptide hormone of white adipose tissue that regulates the release of glucose from hepatocytes via the G-protein/cAMP-protein kinase A pathway [7]. It has been shown that the concentration of asprosin in the plasma of female T2DM patients is higher than in healthy women [8]. A positive correlation was found between asprosin concentration and fasting blood glucose (FBG), HOMA-IR (homeostasis model assessment of insulin resistance), glycated haemoglobin (HbA1c), triglycerides (TG), body mass index (BMI), waist/hip ratio (WHR) and systolic (SBP) as well as diastolic (DBP) blood pressure [8,9,10]. In the research by Jung et al., it has been indicated that asprosin adversely affects the sensitivity of muscle cells to insulin [11]. It is known that the experimental dysfunction of asprosin causes a significant reduction in blood glucose and insulin levels [7]. It has also been demonstrated that pancreatic β cells are a source of asprosin in conditions of hyperlipidaemia, while asprosin induces inflammation, dysfunction and apoptosis of pancreatic β cells, which results in impaired insulin secretion [12].

It has recently been postulated that the use of whole-body cryotherapy (WBC), consisting of short-term (1–3 min) exposure of the whole body to cryogenic temperature (from −110 °C to −160 °C), may be a supplement to pharmacological therapy among obese individuals affected by T2DM [13]. In a recent study, it has been shown that after applying 20 WBC treatments, beneficial changes in the lipid profile occur, consisting in lowering total cholesterol, with a simultaneous decrease in TG, low-density lipoproteins (LDL) and an increase in high-density lipoproteins (HDL), which intensified after 30 WBC treatments. The effects persisted a month after completion of the procedures [14]. In some studies, it was found that the normalised concentration of selected peptide hormones (resistin, visfatin, and irisin) is associated with the metabolism of carbohydrates and lipids under the influence of cryogenic temperatures [15,16,17]. A reduction in the intensity of inflammation after implementing WBC was demonstrated, manifested by a decrease in the concentration of the inflammatory protein (CRP) [16].

Our previous research found that after a series of 20 WBC treatments, an anti-inflammatory effect can be noted and, indirectly, by inducing the secretion of irisin and IL-6, the content of visceral adipose tissue in postmenopausal women without T2DM is also reduced [17]. We have further shown that in postmenopausal women, there is a positive correlation between the concentration of asprosin in the blood and the level of risk factors for metabolic disorders, such as fasting blood glucose, plasma atherogenesis index (AIP) and the leptin/adiponectin index [18]. At the same time, we found that fasting blood asprosin and glucose concentrations are reduced as a result of applying 20 WBC treatments in postmenopausal women with hyperglycaemia but without T2DM [18].

The aim of our present study is to assess the effect of 30 WBCs on asprosin secretion, glucose homeostasis and insulin resistance in postmenopausal women with T2DM.

We hypothesize that after 30 WBC procedures, asprosin concentrations, blood glucose level and insulin resistance in postmenopausal women with T2DM will decrease.

## 2. Materials and Methods

### 2.1. Study Design

Out of 60 volunteers undergoing WBC in Małopolska Cryotherapy Centre from February 2021 to March 2022, 42 people were qualified for the study, 5 of whom resigned without giving a reason. Finally, postmenopausal women (n = 19, 65.89 ± 3.67 years) with type 2 diabetes mellitus (T2DM) and healthy (without T2DM) postmenopausal women (n = 18, 61.56 ± 4.71 years) were included in the study as the control group (CON) (Figure 1).

The examinations included somatic measurements, medical qualification, 30 WBC procedures and quantitative biochemical analyses relevant to glucose homeostasis (fasting blood glucose, glycated haemoglobin, HOMA-IR—homeostasis model assessment of insulin resistance, Quicki index—quantitative insulin sensitivity check index, TyG—triglyceride glucose index, and C-reactive protein).

In the morning, in a fasting state between 6:00 and 8:00 a.m., venous blood was collected using a vacuum system (Becton Dickinson, Franklin Lakes, NJ, USA) from blood vessels in the elbow joint at the following time points:(1)T0: on the day when 1 WBC treatment was performed;(2)T1: the day following the 30 WBC treatments;(3)T2: 2 weeks after completion of the 30th WBC treatment.

Blood was collected after approximately 8 h of sleep (the last meal was eaten 2 h before bedtime). The collected biological material was blinded using a letter-digit code. The analyses were performed in the Laboratory of Biochemistry and Molecular Biology at the University of Physical Education in Krakow.

All the examined women had no contraindications to the use of WBC and had not been subjected WBC treatments for at least 6 months prior to the study [19,20,21].

To exclude the influence of additional uncontrollable factors:(1)The participants were not in physical training and had a low level of physical activity [22].(2)The subjects were asked to maintain their previous physical activity and diet, which was controlled by analysing the results of the International Physical Activity Questionnaire—(IPAQ), the 7-day Physical Activity Recall (7-day PAR) and diaries with the use of the Diet 6.0 program (Institute of Food and Nutrition, Warsaw, Poland) [22,23].(3)The participants did not use any wellness treatments during the WBC protocol.

The research methodology was approved by the Bioethics Committee at the Regional Medical Chamber (10/KBL/OIL/2021; 12 February 2021). The research was carried out in accordance with the Declaration of Helsinki. The volunteers provided their written, informed consent to participate in the study after being informed in writing about the purpose and plan of the study, as well as about the possible inconveniences and risks associated with the study protocol. Participants could withdraw from the study at any stage of its implementation without giving a reason. Before beginning the study, the participants were familiarised with the laboratory conditions, the method of preparation for the procedures and the WBC procedure itself.

### 2.2. Somatic Measurements and Evaluation of Body Composition

Somatic measurements were taken on an empty stomach, between 6.00 and 8.00 a.m., on the day of starting WBC treatments. The results are shown in Table 1.

Body height (BH) was measured to the nearest 1 mm using a stadiometer (Seca 217, Hamburg, Germany). Body mass (BM) was measured in standing position, wearing undergarments (Jawon IOI-353 Body composition analyzer, Gyeongsa, Republic of Korea). Body Mass Index (BMI) was calculated for each participant as body mass (kg)/body height (m)^2^.

Body composition was determined by dual-energy X-ray absorptiometry (DXA) (GE Healthcare Lunar iDXA, Madison, WI, USA), in a supine position, with the arms extended along the trunk, palms facing the thighs. During the measurement, the women wore a sleeveless cotton top and shorts, without plastic, rubber or metal elements. They were asked to remove their glasses and jewellery.

### 2.3. Medical Qualification and Characteristics of the Participants

The medical qualification included anamnesis, physical examination, blood pressure measurement, assessment of laboratory test results and consideration of contraindications to WBC procedures according to the accepted criteria [19,20,21]. In most people with diabetes, the duration of the disease ranges from 1 to 20 years. Diabetic (T2DM) patients were treated with oral antidiabetic drugs, in most cases metformin (n = 15) and sulfonylureas (n = 4). In 3 cases, the drugs also contained sitagliptin, which increases insulin secretion, or dapagliflozin, which causes glycosuria, and 3 people additionally took jardiance, which increases glycosuria. In a few cases, a combination of these drugs was used. Hypertension was treated in 8 patients with T2DM and hypothyroidism in 3 patients. None of the patients who qualified for the programme had morbid complications in the form of peripheral sensation disorders. In the control group (CON), compensated hypertensive disease was found in 4 people and hypothyroidism in 3 individuals. In all cases, both in the T2DM and CON groups, hypertension and hypothyroidism were pharmacologically controlled. Pharmacological treatment was not interfered with during the research programme.

The results of complete blood count, basic biochemical blood parameters and blood pressure of the participants are presented in Table 2.

The T2DM group demonstrated a higher total number of leukocytes (*p* = 0.01), including a higher percentage of neutrophils (*p* = 0.01) than in the CON group, in contrast to the percentage of lymphocytes (*p* < 0.01) and basophils (*p* = 0.04), which were higher in the CON group. The T2DM group had a lower platelet count (*p* = 0.03) compared to CON. The erythrocyte content, haemoglobin concentration and haematocrit did not differ in either of the groups (*p* > 0.05).

The T2DM group showed a higher TG concentration (*p* = 0.01) and AIP value (*p* < 0.01), while the LDL (*p* < 0.01) and total cholesterol (*p* = 0.01) concentrations were lower than in the CON group.

### 2.4. Whole-Body Cryotherapy

Each of the examined persons underwent 30 WBC procedures every day in the afternoon (3:00–5:00 p.m.), excluding Saturdays and Sundays, in Małopolska Cryotherapy Centre from February 2021 to March 2022. WBC treatments were performed under the supervision of a physician and physiotherapist in the Bamet KN-1 stationary cryogenic chamber (Bamet, Wielka Wieś, Poland), cooled with liquid nitrogen, consisting of a vestibule (−60 °C) and a main chamber (−120 °C). During each WBC procedure, the subjects spent 30 s in the atrium and then 3 min in the main chamber. During a single session, a maximum of 4 people could participate in the WBC procedure. The participants walked calmly “in a circle”, one after the other and every 30 s. The direction of the march was changed at a signal. According to the International Classification of Medical Procedures (ICD-9), WBC belongs to physiotherapy measures (93.3950).

Communication with the participants during the treatments was possible through the audio system and thermal windows in the door of the main chamber. The temperature inside the chamber was recorded continuously and the air was dried. The content of oxygen in the air of the cryochamber was maintained at a constant 21–22% and continuously controlled by 2 independent oxygen probes (EurOx.O2 G/E, Kraków, Poland).

The volunteers were asked not to apply cosmetics to the skin before the WBC procedure and to thoroughly remove the sweat so as not to cause frostbite. During the WBC procedure, women wore a sleeveless cotton top and short shorts or leggings that did not put pressure on the skin. The clothes were devoid of metal and other rigid elements. Before beginning the WBC procedure, jewellery and glasses were removed and contact lenses were not used. In addition, each person was equipped with a surgical mask having an additional layer of gauze to protect the nose and mouth, woollen socks covering the ankles and knee pads, gloves, a cap covering the auricles and clogs with wooden soles.

### 2.5. Biochemical Determinations

Blood collected for FBG (K2EDTA and glycolysis inhibitors: sodium fluoride and potassium oxalate) and asprosin (K2EDTA and protease inhibitor: aprotinin 0.6 TIU/1 mL of blood) concentration determinations were centrifuged (RCF 1.000× *g*) for 15 min at 4 °C immediately after collection (MPW-351R, MPW Med. Instruments, Warsaw, Poland). For the determination of insulin, CRP and TG, clotting activator tubes were applied, which, after blood collection, were stored for 20 min at 20–22 °C to clot, and then centrifuged in the above-described conditions. The resulting plasma and serum were stored at −70 °C (ULF 390 Arctiko low-temperature freezer, Esbjerg, Denmark) until analysis.

HbA1c content was determined in whole blood collected with E2DTA using high-performance liquid chromatography (HPLC) via the BIORAD Variant II haematology analyser (Hercules, CA, USA).

CRP concentration was determined via the immune turbidimetric method using the Cardiac C-Reactive Protein (Latex) High Sensitive Test-CRPHS (Roche Diagnostics GmbH, Mannheim, Germany). The detection range of the CRPHS assay was 0.15–20 mg/L, while for the inter-assay, CV < 10%.

FBG and TG concentrations were determined using an enzymatic method, in accordance with the manufacturer’s manual, using reagents dedicated to the Cobas c 701/702 chemistry analyser (Roche Diagnostics GmbH, Mannheim, Germany). The measuring range of the test for glucose (GLUC3) was 2–750 mg/dL and for triglycerides (TRIGL), 8.85–885 mg/dL.

Enzyme-linked immunosorbent assay (ELISA) with absorbance measurement using the Spark^®^ multimode microplate reader (Tecan, Grödig, Austria) was used to determine the concentration of insulin and asprosin. The assays were performed in accordance with the methodology presented by the manufacturer of the reagents: DCM076–8 by DiaMetra (Dia-Metra, Segrate, Italy) and Nori^®^Human Asprosin ELISA Kit GR 111426 (Genorise, Glen Mills, PA, USA), respectively, by reading the results from the standard curve performed during the measurements, verifying the correctness of determinations based on the concentration of the standard sample. The detection range for insulin was 0–200 μIU/mL, the intra-assay coefficient of variation (CV) was ≤5.0% and the inter-assay CV ≤ 10.0%. The detection range for asprosin was 1.5–100 ng/mL, intra-assay CV < 6% and inter-assay CV < 9%.

CRP concentration was determined via the immune turbidimetric method using the Cardiac C-Reactive Protein (Latex) High Sensitive Test-CRPHS (Roche Diagnostics GmbH, Mannheim, Germany). The detection range of the CRPHS assay totalled 0.15–20 mg/L, while for the inter-assay, this value was CV < 10%.

The following indices were calculated: HOMA-IR (homeostasis model assessment of insulin resistance), Quicki (quantitative insulin sensitivity check index) and TyG (triglyceride glucose index), according to formulas:HOMA-IR = insulin (mU/mL) × FBG (mmol/L)/22.5
Quicki = 1/(log insulin (μU/mL) + logFBG (mmol/L)
TyG = lnTG × FBG/2

### 2.6. Statistical Analysis

In the study, 37 people obtained the complete set of results subjected to analysis (T2DM n = 19; CON n = 18); no data were missing from the statistical analysis (Figure 1).

The distribution of results for the analysed variables was checked with the Shapiro-Wilk test. Significance of intergroup differences (T2DM, CON), in the case of single measurements, was assessed using tests for independent samples (the Student’s *t*-test or non-parametric Mann–Whitney U test).

Comparing the impact of WBC treatments on changes in the analysed variables in the contrasted T2DM and CON groups, analysis of variance with repeated measures (ANOVA) was performed. If a significant impact of any of the main factors, i.e., GROUP, WBC and the GROUP × WBC interaction, was found, the significance of differences between specific averages was checked by performing statistical analysis for planned comparisons—*t*-test (post hoc). For changes in the level of biochemical markers after WBC, confidence intervals were determined (95% CI). Effect sizes for ANOVA analysis were calculated using partial eta squared (η^2^) and interpreted as 0.010–0.059 = small, 0.060–0.139 = medium, ≥0.14 = large.

Correlations between variables were determined using Pearson’s or Spearman’s tests. The following correlation assessment was adopted depending on the value of the r correlation coefficient: no correlation if r ≤ 0.19, low correlation if 0.2 ≤ r ≤ 0.39, moderate correlation if 0.40 ≤ r ≤ 0.59, moderately high correlation if 0.6 ≤ r ≤ 0.79, and high correlation if r ≥ 0.8.

The statistical significance of the differences between the compared means was assumed for *p* < 0.05. The STATISTICA 13.3 package (StatSoft, Inc., Tulsa, OK, USA) was used.

The minimum sample size was determined using the G*Power 3.1.9.7 computer program for analysis ANOVA: repeated measures, within–between interaction (3 measures: T0, T1, T2—WBC influence, 2 groups: T2DM, CON—GROUP influence). The sample size was calculated for test power 1-β = 0.90, *p* = 0.05 and effect size η^2^ = 0.06 (medium). For the assumptions adopted in this manner, a total sample size = 36 was obtained, which means 18 people in each of the 2 groups.

## 3. Results

### 3.1. Glucose

There were significant intergroup differences (GROUP; F = 20.82, *p* < 0.01, large effect size), a significant effect of WBC (F = 4.13, *p* = 0.02, medium effect size), and a significant effect of the interaction regarding GROUP × WBC factors (F = 4.19, *p* = 0.02, medium effect size) on FBG (ANOVA) (Table 3).

Post hoc analysis (Table 4) demonstrated a significantly higher FBG concentration before the onset of WBC (T0, *p* < 0.01), after 30 WBCs (T1, *p* < 0.01) and 2 weeks after the end of treatments (T2, *p* < 0.01) in the T2DM group compared to CON.

In the T2DM group, the FBG concentration after 30 WBCs (T1, *p* = 0.04) and 2 weeks after the end of the procedures (T2, *p* < 0.01) was significantly lower than its level recorded prior to the procedure (T0).

In the CON group, there were no significant changes in FBG as a result of WBC (*p* > 0.05).

### 3.2. Glycated Haemoglobin

There were significant intergroup differences (GROUP; F = 6.52, *p* = 0.03, large effect size) and a significant effect of WBC (F = 10.86, *p* < 0.01, large effect size), but no interaction of GROUP × WBC factors (F = 2.24, *p* = 0.14) on the percentage of HbA1c in the blood (ANOVA) (Table 3).

Post hoc analysis (Table 4) showed a significantly higher percentage of HbA_1_c before initiating WBC (T0, *p* = 0.02) in the T2DM group compared to CON, and after 30 WBC treatments (T1) and 2 weeks after completing the procedures (T2), the percentage of HbA1c did not differ significantly in either of the groups, with a tendency towards higher values in the T2DM group (*p* = 0.06).

In the T2DM group, the percentage of HbA_1_c in the blood decreased significantly after 30 WBC treatments (T1, *p* < 0.01) and 2 weeks after the procedures (T2, *p* = 0.01) compared to their baseline value (T0).

In the CON group, there were no significant changes in the percentage of HbA1c in the blood as a result of WBC (*p* > 0.05) (Table 4).

### 3.3. Insulin

There were significant between-group differences (GROUP; F = 12.27, *p* < 0.01, large effect size) and a significant effect of WBC (F = 4.67, *p* = 0.01, medium effect size), without the influence of the interaction of GROUP × WBC factors (F = 0.23, *p* = 0.80) on serum insulin concentration (ANOVA) (Table 3).

Significantly higher insulin concentrations were demonstrated (post hoc) before beginning WBC (T0, *p* < 0.01), after 30 WBCs (T1, *p* < 0.01) and 2 weeks after the end of the procedures (T2, *p* < 0.01) in the T2DM group compared to CON (Table 4).

In the T2DM group, the insulin concentration 2 weeks after the end of WBC procedures (T2, *p* = 0.04) was significantly lower than prior to their implementation (T0).

In the CON group, there was a tendency towards lower insulin levels 2 weeks after completing WBC procedures (T2, *p* = 0.07).

### 3.4. HOMA-IR and Quicki Indices

For the values of HOMA-IR and Quicki indices, significant intergroup differences were found (GROUP; F = 11.73, *p* < 0.01, large effect size and F = 17.89, *p* < 0.01, large effect size, respectively). A significant effect of WBC (F= 8.01, *p* < 0.01, large effect size and F = 5.76, *p* = 0.01, large effect size), without the interaction of GROUP × WBC factors (F = 0.87, *p* = 0.42 and F = 0.09, *p* = 0.92) was found on the value of both indices (ANOVA) (Table 3).

Post hoc analysis (Table 4) allowed to show significantly (*p* < 0.01) higher HOMA-IR and significantly (*p* < 0.01) lower Quicki values in the T2DM group before initiating WBC (T0), after 30 WBCs (T1) and after 2 weeks after the end of treatments (T2), compared to CON.

In the T2DM group, 2 weeks after the completion of WBC (T2) procedures, the HOMA-IR value was significantly lower (*p* = 0.01), while the Quicki index was significantly higher (*p* = 0.02) compared to the value before the WBC treatment (T0).

In the CON group, 2 weeks after the end of WBC treatments (T2), a tendency towards decreasing HOMA-IR (*p* = 0.08) and increasing Quicki value (*p* = 0.05) was found.

### 3.5. TyG Index

There were significant intergroup differences in TyG values (GROUP; F = 18.06, *p* < 0.01, large effect size). However, there was no significant effect of WBC (F = 0.53, *p* = 0.59) or GROUP × WBC factor interaction (F = 0.52, *p* = 0.60) on TyG (ANOVA) (Table 3).

Post hoc analysis (Table 4) exhibited significantly higher TyG values in the T2DM group before starting WBC (T0, *p* < 0.01), after 30 WBC treatments (T1, *p* < 0.01) and 2 weeks after ending the procedures (T2, *p* < 0.01) compared to CON.

### 3.6. Asprosin

A significant effect of WBC on plasma asprosin concentration was found (F = 6.03, *p* < 0.01, large effect size). However, there were no significant intergroup differences (GROUP; F = 0.76, *p* = 0.39) or influence of the GROUP × WBC factor interaction (F = 0.67, *p* = 0.51) on the concentration of this hormone (ANOVA) (Table 3).

Two weeks after the WBC (T2) procedures, a post hoc trend towards a decrease in the concentration of asprosin in the T2DM group (*p* = 0.06) and a significant decrease in the CON group (*p* = 0.03) were noted (Table 4).

### 3.7. C-Reactive Protein

A significant effect of WBC on CRP in the serum was found (F = 4.95, *p* = 0.01, large effect size). However, there were no significant intergroup differences (GROUP; F = 1.85, *p* = 0.18) or influence of the GROUP × WBC factor interaction (F = 0.03, *p* = 0.98) on the concentration of this protein (ANOVA) (Table 3).

It was found (post hoc) that in the T2DM group, 2 weeks after the end of the WBC procedures (T2), the CRP concentration decreased significantly (*p* = 0.04) compared to the baseline value (T0) (Table 4).

### 3.8. Correlations

A significant (*p* < 0.05) positive correlation between the asprosin concentration and FBG (r = 0.91) and HOMA-IR value (r = 0.89) was found in the whole group. There was also a significant (*p* < 0.05) positive correlation between the insulin concentration and percentage of body fat in the trunk (r = 0.83) and abdominal areas (r = 0.81), as well as between insulin and HOMA-IR (r = 0.89) and the value of the Quicki index (r = −0.96). Changes in insulin levels after 30 WBCs (T1-T0) and 2 weeks after the end of the 30 WBC series correlated positively (*p* < 0.05) with HOMA-IR (r = 0.98 and r = 0.99, respectively) and negatively (*p* < 0.05) with the value of the Quicki index (r = −0.96 and r = −0.97, respectively).

In the T2DM group, there was a significant (*p* < 0.05) positive correlation between the asprosin concentration and FBG (r = 0.98) and HOMA-IR value (r = 0.98). There was also a significant (*p* < 0.05) negative correlation between the asprosin concentration and lean body mass (r = −0.95), as well as a significant (*p* < 0.05) positive correlation between the change in insulin concentration and the change in HOMA-IR (r = 0.98). A significant (*p* < 0.05) negative correlation was further noted between the change in insulin concentration and the change in the Quicki index value, calculated between T2 and T0.

In the CON group, a significant (*p* < 0.05) positive correlation (r = 0.99) was observed between the change in the asprosin concentration after 30 WBC treatments and the change in HOMA-IR after 30 WBCs (T1-T0).

## 4. Discussion

In our research, it has been shown that the use of WBC therapy can be an adjuvant therapy in the treatment of T2DM. We have demonstrated that after 30 WBCs, the level of glucose in the blood and the content of glycated haemoglobin decrease, and these changes persist for 2 weeks after completing the procedures. This is also associated with a decrease in the concentration of insulin in the blood, improvement in the values of insulin resistance indicators (HOMA-IR, Quicki) and decreased C-reactive protein secretion. This further demonstrates improvement in insulin sensitivity and reduction in chronic inflammation accompanying T2DM. We found a decreasing trend in blood asprosin levels that correlate with glucose levels and HOMA-IR in postmenopausal women with T2DM. Contrary to the T2DM group, for the control, 2 weeks after 30 WBCs, the concentration of asprosin in the blood significantly decreased, which was accompanied by a tendency towards a decrease in the concentration of insulin, HOMA-IR and a simultaneous increase in the value of the Quicki index, which proves improved insulin sensitivity in peripheral tissues.

In earlier studies, it has been shown that an important factor related to the level of asprosin is elevated blood glucose level [18]. Women with hyperglycaemia but without T2DM demonstrated higher levels of asprosin than those with normal glycaemia [18].

In the current study, we have shown that the fasting blood asprosin concentration before the onset of WBC, both in the entire group of subjects and in that with T2DM, indicates a strong positive correlation with FBG and HOMA-IR, although the concentration of asprosin in our research was comparable in the T2DM group and control without T2DM. It has previously been noted that fasting plasma asprosin concentration is significantly higher in patients with newly diagnosed T2DM compared to the control group [9,24]. However, our study involved people suffering from T2DM for a year up to even 20 years.

Similar to our research, other researchers, in patients with T2DM, also showed a positive correlation between the concentration of asprosin and HOMA-IR, FBG, and also with TG [9]. However, as in our trial, there was no correlation between asprosin and HbA_1_c [9]. In another study involving healthy women and patients with T2DM, as well as those with polycystic ovary syndrome (PCOS), it was indicated that the concentration of asprosin in the plasma of T2DM and PCOS patients is comparable and, at the same time, in both groups, is higher than in healthy women. In these groups of patients, plasma asprosin levels correlated positively with HOMA-IR and HbA_1_c [8]. The results of our own study and those of other researchers indicate a close relationship between the secretion of asprosin and disorders of glucose homeostasis as well as insulin resistance [8,9,18,24].

As a result of excessive adipose tissue growth in the body, the secretion of asprosin, which is an adipocytokine, may be disturbed, and, consequently, gluconeogenesis may be stimulated while hepatic insulin resistance increases [9]. Some authors have demonstrated that the concentration of asprosin shows a positive correlation with BMI and the waist/hip ratio (WHR) [8,10]. It was found that the concentration of asprosin increases gradually in groups separated on the basis of increasing BMI, i.e., low weight (BMI < 18.5), normal weight (BMI: 18.5–24.9), overweight (BMI: 25.0–29.9), and obese class I (BMI: 30.0–34.9), II (BMI: 35.0–39.9) and III (BMI ≥ 40.0) [25]. In our study, based on the correlation analysis, we found that in the T2DM group, the asprosin concentration is lower in people with greater lean body mass (LBM), but we did not find any relationship between asprosin concentration and BMI. The groups we studied (T2DM, CON) did not differ in terms of BMI or LBM, which may explain the lack of significant intergroup differences in asprosin concentration.

Physiologically, the concentration of asprosin in the plasma shows daily fluctuations. The highest values are present in a fasting state, and the secretion of asprosin is regulated by negative feedback, where the role of the suppressor is played by the increased concentration of glucose in the blood [7]. On the basis of changes in asprosin concentration as a response to the oral glucose tolerance test (OGTT), it has been shown that in T2DM patients, the secretion of asprosin is disturbed and not regulated by glucose concentration. This may potentially be a mechanism of disease development [26]. In the OGTT test, the concentration of asprosin in healthy subjects was significantly reduced 2 h after administration of 75 g of glucose, but there was no such effect in subjects with T2DM [26]. This is also confirmed by the high, positive correlation between the concentration of asprosin and FBG, which we obtained in the T2DM group in our study, proving the lack of inhibition of asprosin secretion by glucose in patients. At the same time, we found a high correlation between the concentration of asprosin and HOMA-IR in the T2DM group, which proves the participation of this adipocytokine in the development of insulin resistance.

In research involving subjects with normoglycaemia, prediabetes and T2DM, a higher concentration of asprosin was found in the group of patients compared to subjects with normal glycaemia; however, the highest concentration of this hormone was found in patients with prediabetes [24]. The control group in our study, although it consisted of people without diagnosed T2DM, included eight people with a borderline level of glycated haemoglobin (5.7–5.8%), which may indicate pre-diabetes. People in the CON group also had higher LDL levels than in the T2DM group. This may be related to the increased level of asprosin in the CON group and may explain the lack of significant differences in the baseline level of this hormone in the studied groups.

In previous research, we have shown that repeated, whole-body exposure to cryogenic temperatures reduces the level of asprosin in the blood of postmenopausal women [18]. At the same time, we found that a decrease in asprosin concentration after 20 WBCs is correlated with a decrease in blood glucose levels. Comparing the results obtained for women with hyperglycaemia (prediabetes) to the results of those with normoglycaemia, we note a decrease in blood asprosin level after WBC, but only in the group with hyperglycaemia, although we did not find a significant relationship between the change in asprosin concentration after WBC and the baseline concentration of blood glucose. Nonetheless, after applying 20 WBCs, we did not observe any significant changes in insulin levels or insulin resistance index (HOMA-IR) in any of the study groups [18].

Based on these results, we have now applied 30 WBCs, including postmenopausal women with T2DM in the study [18]. We used WBC treatments every day, in six series of five treatments a week, excluding Saturdays and Sundays. This is the first study in which cryogenic temperatures were used as a factor inducing the normalisation of glucose metabolism among people with T2DM. In our current study, after 30 WBCs, we achieved a reduction in blood glucose and HbA_1_c in the T2DM group, which was also maintained for 2 weeks after completing the treatments. These changes were accompanied by a decrease in insulin levels associated with improvement in insulin resistance indices. Throughout the study, participants maintained their diet and physical activity to eliminate the potential impact of these factors on glycaemic levels, which were controlled. Similar to our research, after 10 WBCs, improvement in glycaemia as well as a decrease in insulin concentration and insulin resistance were obtained in men around the age of 50 [27]. However, this is not confirmed by other studies in which the use of the 20 WBC series did not affect the level of glycaemia in obese or healthy men [15,28]. Differences in the effects of using WBC may result, among others, from different procedures for conducting WBC treatments (number of treatments, duration of the procedure, treatment temperature). Even a different number of people participating in the WBC procedure at the same time affects temperature distribution in the cryochamber, which, in the case of a constant single treatment time, may affect the results [29]. In our study, the treatment time was constant and equalled 3 min, but the WBC procedure was always attended by four people, and the conditions inside the cryochamber were continuously controlled in order to minimise changes in treatment conditions.

It seems that the changes in asprosin concentration obtained in our study were independent of the changes in glucose concentration that occurred as a result of introducing cryogenic temperatures. The reduction in glycaemia after 30 WBC sessions occurred only in the T2DM group and persisted for 2 weeks after the procedures. However, in both groups of postmenopausal women, there was a decrease in the blood asprosin concentration only 2 weeks after the end of WBC procedures, although in the T2DM group, this change was not statistically significant, unlike the change in the control group. This may be due to the higher production of asprosin in the T2DM group and/or thermal insulation, caused by the higher percentage of adipose tissue in postmenopausal women with T2DM, especially in the abdominal area, which is the main source of asprosin secretion [7].

The glycaemic-normalising effect of cryogenic temperatures is directly related to the mechanisms of thermoregulation during the procedure. The primary physiological mechanism, preventing the reduction of core temperature in response to cryogenic temperatures, is the stimulation of the sympathetic nervous system and vasoconstriction in the skin and subcutaneous tissue. It may also induce shivering thermogenesis. During the second phase, following the procedure, vasodilation and an increase in perfusion take place [13,19].

Increased intracellular glucose uptake is associated with the mechanism of thermoregulation induced in response to cold. In animal models, prolonged exposure to cold (4–10 °C) or norepinephrine infusion has been associated with an increased glucose uptake into the white adipose tissue (WAT) of rodents in both the basal- and insulin-stimulated state, and these effects persisted upon returning to thermoneutrality. It can be speculated that the increased glucose uptake is used as fuel for the increased lipolysis energy demand during cold exposure or restoration of TG lipid pool following cold exposure [30]. In persons exposed to cryogenic temperatures, the same effect may occur. There was a decrease in skin and core temperature, an increase in norepinephrine concentration, and an increase in oxygen uptake per minute as well as energy expenditure, additionally accompanied by shivering thermogenesis [31,32].

In studies on rodents and humans, shiver-induced glucose uptake has been noted in the skeletal muscles, independently of insulin secretion, which may be due to adipomyokine irisin, released as a result of muscle cell contractions [30,33,34]. Irisin affects the browning of WAT by inducing the expression of uncoupling protein 1 (UCP-1) and upregulates the expression of UCP-3 in myocytes, thereby increasing energy expenditure caused by non-shivering thermogenesis [30]. Irisin affects faster depletion of cellular energy substrates, glucose homeostasis and improves insulin sensitivity [33,34]. It has been demonstrated that exogenous irisin stimulates the activation of lipoprotein lipase and triglyceride lipolysis, improves lipid metabolism and glucose uptake by cells, and reduces lipid synthesis in mice [35,36]. At the same time, it was found that irisin regulates the level of asprosin in the serum and reduces the level of LDL, TG, glucose and leptin, which indicates an important role of irisin in the regulation of metabolism, counteracting insulin resistance and its anti-inflammatory effect [37].

Whole-body cryotherapy increased irisin levels in both lean and obese men, as well as in postmenopausal women [16,17,27,38]. In our previous research, a significant impact was indicated with regard to WBC on increasing the concentration of irisin in the blood during the initial phase of WBC treatments, i.e., after single exposure and after 10 WBC treatments, both in healthy people and in patients with metabolic syndrome [17], which may initiate a lowering of glucose levels. However, after 20 WBCs, the irisin level stabilised at the baseline level [17,18]. Unfortunately, in this study, we did not evaluate the changes in irisin concentration among postmenopausal women with T2DM induced by whole-body cryotherapy, which makes it impossible to consider the role of irisin in the mechanism of the obtained changes in glucose concentration and constitutes a limitation of our study.

According to literature on the subject, whole-body cryotherapy has anti-inflammatory effects, and could potentially influence CRP levels [39,40,41]. A significant decrease in the concentration of pro-inflammatory interleukins, such as tumour necrosis factor α (TNF-α) and interleukin 6 (IL-6), as well as an increase in the concentration of anti-inflammatory interleukin 10 (IL-10), were found after just 10 WBC procedures [39]. This effect is also beneficial within the context of improving insulin resistance, because chronically elevated levels of TNF-α and IL-6 impair the ability of insulin to stimulate intracellular glucose translocation with the use of insulin-dependent glucose transporter 4 (GLUT 4), mainly in myocytes [30]. Some researchers have also reported a decrease in CRP after 10 sessions of WBC [16,39]. However, in some trials, these results have not been confirmed. For example, there were no significant changes in CRP concentration after 10 WBCs in 20-year-old non-obese men, regardless of their level of physical fitness [38]. However, in obese 40-year-old males, in the group demonstrating low and high levels physical fitness, CRP concentration was significantly lower than before beginning cryotherapy after 10 WBC sessions [38]. On the other hand, in contrast to the lack of changes in CRP levels in postmenopausal women [17], blood CRP concentrations in obese men decreased after 20 WBCs [17,42]. In our current research, the reduction of insulin resistance in postmenopausal women with T2DM was associated with the anti-inflammatory effect of cryotherapy, expressed by a decrease in blood CRP levels 2 weeks after the end of the treatments.

### Limitation of the Study

Our study included 37 volunteers, all of whom were menopausal women. It is advisable to carry out studies with more volunteers with T2DM, both women and men of all ages, in the future. The mechanism of reducing insulin resistance as a result of whole-body cryotherapy has not been fully elucidated. Our research did not explain this mechanism. A clear determination of whether whole-body cryotherapy can be an adjunctive therapy in the treatment of T2DM and an effective prevention of insulin resistance requires further, more in-depth research. In future research, the possible impact of cryogenic temperatures on the expression of microRNAs involved in the regulation of metabolism in T2DM and energy processes should be considered [43,44].

## 5. Conclusions

In conclusion, 30 WBC treatments cause a reduction in glucose, insulin resistance and chronic inflammation among postmenopausal women with T2DM. Asprosin concentration in the T2DM group positively correlates with FBG and HOMA-IR and shows a downward trend following WBC procedures. In postmenopausal women without T2DM, the concentration of asprosin decreases significantly after WBC, and the decrease correlates with a drop in insulin resistance.

## Figures and Tables

**Figure 1 biomolecules-13-01602-f001:**
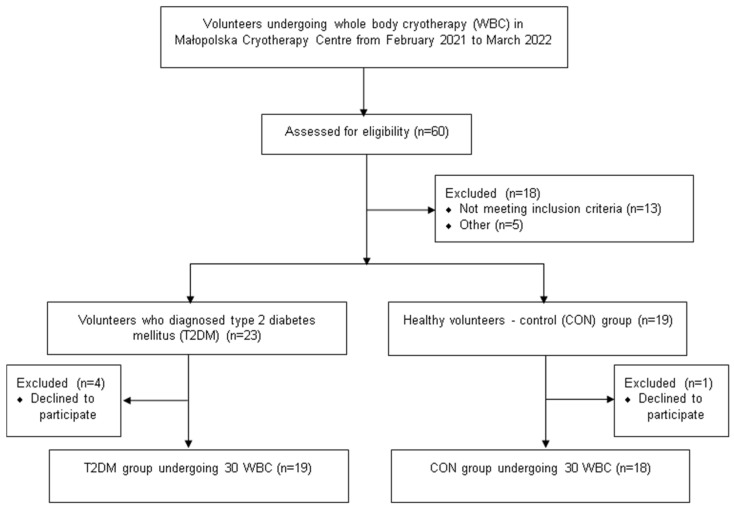
Flow chart of the procedure of participants’ qualifications.

**Table 1 biomolecules-13-01602-t001:** Age and somatic structure of the participants.

Variable	T2DM	CON	*p*-Value
Age (years)	65.89 ± 3.67	61.56 ± 4.71	<0.01
Body Mass (kg)	78.77 ± 9.94	73.94 ± 8.62	0.13
BMI (kg/m^2^)	31.18 ± 4.51	28.96 ± 5.31	0.08
Lean Body Mass (kg)	43.95 ± 4.62	43.89 ± 3.72	0.96
Total Body Fat (kg)	34.81 ± 6.65	34.42 ± 13.13	0.28
Total Body Fat (%)	43.92 ± 4.28	40.13 ± 5.58	0.03
Trunk Fat (%)	48.66 ± 5.26	42.97 ± 6.75	0.01
Android Fat (%)	50.78 ± 6.80	45.02 ± 8.30	0.02
Gynoid Fat (%)	44.55 ± 4.52	43.13 ± 5.58	0.40
Android/Gynoid (A/G)	1.12 ± 0.12	1.03 ± 0.13	0.02

**Abbreviations:** Values are means ± SD; SD, standard deviation; BMI, body mass index; *p* < 0.05, significant differences between type 2 diabetes mellitus group (T2DM) vs. healthy women—control group (CON) (*t*-test or Mann–Whitney U test).

**Table 2 biomolecules-13-01602-t002:** Medical qualification.

Variable	T2DM	CON	*p*-Value
Erythrocytes (10^6^/µL)	4.51 ± 1.96	4.40 ± 0.26	0.28
Haemoglobin (g/dL)	13.76 ± 1.08	13.31 ± 0.93	0.18
Haematocrit (%)	39.88 ± 2.86	38.88 ± 3.12	0.32
Platelets (10^3^/µL)	228.94 ± 52.32	267.94 ± 52.22	0.03
Leukocytes (10^3^/µL)	6.82 ± 1.96	5.44 ± 1.16	0.01
Neutrophils (%)	55.34 ± 7.84	47.72 ± 8.64	0.01
Lymphocytes (%)	32.29 ± 6.37	40.17 ± 8.05	<0.01
Monocytes (%)	8.35 ± 1.35	8.59 ± 1.88	0.66
Eosinophils (%)	3.26 ± 2.38	2.64 ± 1.34	0.59
Basophils (%)	0.75 ± 0.38	0.88 ± 0.22	0.04
Creatinine (µmol/L)	65.47 ± 11.33	71.48 ± 10.58	0.11
TCHOL (mmol/L)	4.50 ± 1.24	5.70 ± 1.19	0.01
LDL (mmol/L)	2.16 ± 1.12	3.29 ± 0.98	<0.01
HDL (mmol/L)	1.59 ± 0.52	1.89 ± 0.39	0.06
TG (mmol/L)	1.63 ± 0.62	1.16 ± 0.43	0.01
AIP (log_10_TG/HDL)	−0.01 ± 0.22	−0.23 ± 0.20	<0.01
SBP (mmHg)	127.27 ± 7.54	126.56 ± 12.21	0.96
DBP (mmHg)	79.09 ± 3.02	78.13 ± 6.29	0.64

**Abbreviations:** Values are means ± SD; SD, standard deviation; TCHOL, total cholesterol; LDL, low-density lipoproteins; HDL, high-density lipoproteins; TG, triglycerides; AIP, atherogenic index of plasma; SBP, systolic blood pressure; DBP, diastolic blood pressure; *p* < 0.05, significant differences: type 2 diabetes mellitus group (T2DM) vs. healthy women—control group (CON) (*t*-test or Mann–Whitney U test).

**Table 3 biomolecules-13-01602-t003:** Assessing the effect of whole-body cryotherapy treatments on changes in glucose metabolism indices among control postmenopausal women as well as those with type 2 diabetes mellitus (T2DM)—analysis of variance with repeated measures (ANOVA).

Variable	*p*	F	Effect Size η^2^	Power Test 1-β
	Group effect			
FBG (mmol/L)	<0.01	20.82	0.37	0.99
HbA1c (%)	0.03	6.52	0.45	0.61
Insulin (µIU/mL)	<0.01	12.27	0.29	0.92
HOMA-IR	<0.01	11.73	0.72	1.00
Quicki index	<0.01	17.89	0.36	0.98
TyG index	<0.01	18.06	0.35	0.99
Asprosin (ng/mL)	0.39	0.76	0.02	0.13
CRP (µg/mL)	0.18	1.85	0.05	0.26
	WBC effect			
FBG (mmol/L)	0.02	4.13	0.11	0.71
HbA1c (%)	<0.01	10.86	0.58	0.97
Insulin (µIU/mL)	0.01	4.67	0.13	0.76
HOMA-IR	<0.01	8.01	0.20	0.95
Quicki index	0.01	5.76	0.15	0.85
TyG index	0.59	0.53	0.02	0.13
Asprosin (ng/mL)	<0.01	6.03	0.16	0.87
CRP (µg/mL)	0.01	4.95	1.33	0.79
	Interaction GROUP × WBC effect			
FBG (mmol/L)	0.02	4.19	0.11	0.72
HbA1c (%)	0.14	2.24	0.22	0.39
Insulin (µIU/mL)	0.80	0.23	0.01	0.80
HOMA-IR	0.42	0.87	0.03	0.19
Quicki index	0.92	0.09	<0.01	0.06
TyG index	0.60	0.52	0.02	0.13
Asprosin (ng/mL)	0.51	0.67	0.02	0.16
CRP (µg/mL)	0.98	0.03	<0.01	0.05

**Abbreviations:** WBC, whole-body cryotherapy; FBG, fasting blood glucose; HbA1c, glycated haemoglobin; HOMA-IR, homeostasis model assessment of insulin resistance; Quicki index, quantitative insulin sensitivity check index; TyG, triglyceride glucose index; CRP, C-reactive protein; *p* < 0.05, statistically significant effect; η^2^, partial eta squared.

**Table 4 biomolecules-13-01602-t004:** Changes in glucose metabolism indices in control postmenopausal women (CON) as well as those with type 2 diabetes (T2DM) as an effect of whole-body cryotherapy treatments.

	T0	T1	T2	ΔT1-T0	ΔT2-T0
	Mean ± SD	Mean ± SD	Mean ± SD	Mean (95% CI)	Mean (95% CI)
T2DM					
FBG (mmol/L)	6.97 ± 1.56 ^#^	6.68 ± 1.53 ^#^	6.34 ± 1.40 ^#^	−0.30 (−0.68; 0.08) *	−0.64 (−1.11; −0.17) *
HbA1c (%)	6.42 ± 0.78 ^#^	6.24 ± 0.68	6.20 ± 0.80	−0.18 (−0.32; −0.04) *	−0.40 (−1.00; 0.20) *
Insulin (µIU/mL)	18.06 ± 7.60 ^#^	16.99 ± 8.44 ^#^	14.67 ± 6.58 ^#^	−0.69 (−3.70; 2.31)	−3.39 (−6.25; −0.52) *
HOMA-IR	5.80 ± 3.27 ^#^	5.32 ± 3.61 ^#^	4.38 ± 2.97 ^#^	−0.38 (−1.19; 0.43)	−1.43 (−2.29; −0.56) *
Quicki index	0.30 ± 0.03 ^#^	0.31 ± 0.03 ^#^	0.32 ± 0.03 ^#^	0.00 (−0.00; 0.01)	0.02 (0.01; 0.02) *
TyG index	9.02 ± 0.49 ^#^	8.94 ± 0.54 ^#^	8.93 ± 047 ^#^	−0.08 (−0.23; 0.07)	−0.09 (−0.27; 0.09)
Asprosin (ng/mL)	3.34 ± 2.58	2.96 ± 2.66	2.83 ± 2.59	−0.38 (−0.92; 0.16)	−0.51 (−1.05; 0.02)
CRP (µg/mL)	2.66 ± 1.58	2.01 ± 1.06	1.80 ± 0.99	−0.59 (−1.35; 0.16)	−0.86 (−1.72; −0.01) *
CON					
FBG (mmol/L)	5.10 ± 0.37	5.10 ± 0.48	5.10 ± 0.36	0.01 (−0.13; 0.15)	0.00 (−0.14; 0.15)
HbA1c (%)	5.56 ± 0.18	5.43 ± 0.19	5.45 ± 0.21	−0.13 (−0.20; −0.05)	−0.15 (−0.31; 0.01)
Insulin (µIU/mL)	11.02 ± 5.44	9.29 ± 3.65	8.27 ± 2.58	−1.73 (−4.90; 1.43)	−2.75 (−5.97; 0.47)
HOMA-IR	2.62 ± 1.33	2.35 ± 1.29	1.98 ± 0.70	−0.27 (−0.98; 0.44)	−0.64 (−1.31; 0.44)
Quicki index	0.34 ± 0.03	0.34 ± 0.02	0.35 ± 0.02	0.01 (−0.01; 0.02)	0.01 (−0.00; 0.02) *
TyG index	8.39 ± 0.43	8.39 ± 0.40	8.32 ± 0.36	0.02 (−0.13; 0.18)	−0.04 (−0.20; 0.11)
Asprosin (ng/mL)	2.94 ± 2.00	2.27 ± 2.17	1.87 ± 1.44	−0.68 (−1.69; 0.34)	−1.08 (−2.02; −0.14) *
CRP (µg/mL)	2.11 ± 2.11	1.46 ± 0.91	1.42 ± 0.80	−0.65 (−1.48; 0.18)	−0.69 (−1.54; 0.16)

**Abbreviations:** SD, standard deviation; CI, confidence interval; ΔT1-T0, difference after 30 WBCs (T1), compared to pre 1 WBC (T0); ΔT2-T0, difference 2 weeks (T2) after the 30WBC, compared to pre 1 WBC (T0); T2DM, type 2 diabetes mellitus postmenopausal women group; CON, healthy postmenopausal women—control group; FBG, fasting blood glucose; HbA1c, glycated haemoglobin; HOMA-IR, homeostasis model assessment of insulin resistance; Quicki index, quantitative insulin sensitivity check index; TyG, triglyceride glucose index; CRP, C-reactive protein; * statistically significant differences compared to values pre 1 WBC (*p* < 0.05); ^#^ statistically significant differences between groups (*p* < 0.05).

## Data Availability

The data analysed during the study are available from the corresponding author on reasonable request.

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
