# Peer review of "Whole-Body Cryotherapy Improves Asprosin Secretion and Insulin Sensitivity in Postmenopausal Women–Perspectives in the Management of Type 2 Diabetes"

_biomolecules, 2023, doi:10.3390/biom13111602_

Round 1
Reviewer 1 Report
Comments and Suggestions for Authors
The sample size does not appear to have sufficient power to demonstrate the differences between the study groups. More information is required concerning the calculation of the sample size, which is indicated as 36; however, taking into account the objective, it probably should correspond to 36 participants per group. This point is important for the interpretation of the results and the conclusions. I ask the authors to comment.
In Table 1, it is noted that there are no differences between the groups in Gynoid Fat (%), while the group with T2DM is higher in Android Fat (%); however, A/G is significantly higher in the control group. Please review.
Table 2 includes complete blood count. Some comment is required concerning the differences presented between groups. On the other hand, there is no report of levels of glucose, HOMA and asprosin in each group.
In the same Table 2, significant differences are observed between the control and T2DM groups in various fractions of lipids. As the same authors note, there is an association between lipids and asprosin, and this may have affected the final results. Please include a comment.
In Table 3, the Power test 1-b is low, particularly in the section on Interaction GroupxWBC effect.
In Table 4, values of dT2-T0 are marked significant for the Quicki index, but 95%CI does not appear to correspond. Please verify.
In Table 4, the average value of HbA1c according to the deviation appears to be over 5.7%, which suggests that some participants may present prediabetes, which would have influenced the interpretation of the results. Please comment.
The authors feel that the evaluation of 19 patients with T2DM is sufficient to arrive at the conclusion: “In our study, it has been shown that multiple WBC treatments can be a therapy supporting the treatment of T2DM and may be a preventative measure for insulin resistance. The results of the research may be used in clinical practice in diabetology”. Is this justified?
Reviewer 2 Report
Comments and Suggestions for Authors
The manuscript by Wiecek et al. determined the effect of whole-body cryotherapy (WBC) on glucose homeostasis, insulin sensitivity, and inflammation in postmenopausal women with T2DM (T2DM) compared to postmenopausal women without T2DM (CON). The authors reported reduced glucose levels, insulin resistance, and inflammation in the T2DM group after 30 WBCs. They also observed the decreased levels of asprosin, which functions to increase the glucose level by modulating hepatic glucose release, in both T2DM and CON groups. Additionally, the decrease in the asprosin levels positively correlated to decreased insulin resistance. The manuscript is well-written with a comprehensive introduction and discussion and results are clearly presented. As a result, the manuscript can be considered for publication in Biomolecules with minor revision.
Minor comments
- The irisin levels can offer great insight into the mechanism by which WBC affects glucose homeostasis. The authors did not evaluate the irisin levels in the current study and discussed it as a limitation of the study. The authors could consider including in the discussion results from their J. Clin. Med. 2019 paper, reporting the change of irisin levels after 20 WBCs.
- Since the study in its current form did not explore the mechanisms by which the WBC reduces insulin resistance. As a result, the statement in line 29 and line 565 that WBC treatment “can/may be a preventative measure for insulin resistance” is overstated.
Author Response
Thank you for your comments. The text has been corrected in accordance with the reviewer's comments. The changes are marked in the manuscript in red. The following is a point-by-point response to the comments:
- The irisin levels can offer great insight into the mechanism by which WBC affects glucose homeostasis. The authors did not evaluate the irisin levels in the current study and discussed it as a limitation of the study. The authors could consider including in the discussion results from their J. Clin. Med. 2019 paper, reporting the change of irisin levels after 20 WBCs.
In the work: J. Clin. Med. 2019 (reference 18), no changes in irisin concentration were demonstrated after 20 WBC sessions, but in the work from J. Clin. Med. 2020 (item 17 on reference list), an increase was shown in irisin concentration after a single WBC treatment and after 10 WBC sessions.
The following fragment was added to the ‘Discussion’ section: "Our previous research allows to indicate the significant impact of WBC on increasing the concentration of irisin in the blood at the initial phase of WBC treatments, i.e. after single exposure and after 10 WBC sessions, both in healthy people and in patients with metabolic syndrome [17], which may initiate a decrease in glucose concentration. However, after 20 WBCs, the irisin level stabilised at the baseline level [17,18]” (lines 542-547).
- Since the study in its current form did not explore the mechanisms by which the WBC reduces insulin resistance. As a result, the statement in line 29 and line 565 that WBC treatment “can/may be a preventative measure for insulin resistance” is overstated.
Line 29 and a fragment from the ‘Conclusion’ have been deleted: "In our study, it has been shown that multiple WBC treatments can be a therapy supporting the treatment of T2DM and may be a preventative measure for insulin resistance. The results of the research may be used in clinical practice in diabetes”.
The ‘Limitations’ of the study were supplemented with the text: "A clear determination of whether whole-body cryotherapy can be adjunctive in the treatment of T2DM and an effective prevention of insulin resistance requires further, more in-depth research" (lines 575-577).
Round 2
Reviewer 1 Report
Comments and Suggestions for Authors
The authors made the requested modifications.